# Assessment of antibody dynamics and neutralizing activity using serological assay after SARS-CoV-2 infection and vaccination

**Toshihiro Takahashi[1], Tomohiko Ai[2], Kaori Saito[2], Shuko Nojiri[3], Maika Takahashi[1], Gene Igawa[1], Takamasa Yamamoto[1], Abdullah Khasawneh[2], Faith Jessica Paran[4], Satomi Takei[2], Yuki Horiuchi[2], Takayuki Kanno[5], Minoru Tobiume[5], Makoto Hiki[6,7], Mitsuru Wakita[1], Takashi Miida[2], Atsushi Okuzawa[4,8], Tadaki Suzuki[5], Kazuhisa Takahashi[4,9], Toshio Naito[4,10], Yoko Tabe[2,4]***

1 Department of Clinical Laboratory, Juntendo University Hospital, Tokyo, Japan, 2 Department of Clinical Laboratory Medicine, Juntendo University Graduate School of Medicine, Tokyo, Japan, 3 Medical Technology Innovation Center, Juntendo University, Tokyo, Japan, 4 Department of Research Support Utilizing Bioresource Bank, Juntendo University Graduate School of Medicine, Tokyo, Japan, 5 Department of Pathology, National Institute of Infectious Diseases, Tokyo, Japan, 6 Department of Emergency Medicine, Juntendo University Faculty of Medicine, Tokyo, Japan, 7 Department of Cardiovascular Biology and Medicine, Juntendo University Faculty of Medicine, Tokyo, Japan, 8 Department of Coloproctological Surgery, Juntendo University Graduate School of Medicine, Tokyo, Japan, 9 Department of Respiratory Medicine, Juntendo University Graduate School of Medicine, Tokyo, Japan, 10 Department of General Medicine, Juntendo University Graduate School of Medicine, Tokyo, Japan

* tabe@juntendo.ac.jp

**Data Availability Statement:** All relevant data are within the manuscript and the accompanying tables and figures.

## Abstract

The COVID-19 antibody test was developed to investigate the humoral immune response to SARS-CoV-2 infection. In this study, we examined whether S antibody titers measured using the anti-SARS-CoV-2 IgG II Quant assay (S-IgG), a high-throughput test method, reflects the neutralizing capacity acquired after SARS-CoV-2 infection or vaccination. To assess the antibody dynamics and neutralizing potency, we utilized a total of 457 serum samples from 253 individuals: 325 samples from 128 COVID-19 patients including 136 samples from 29 severe/critical cases (Group S), 155 samples from 71 mild/moderate cases (Group M), and 132 samples from 132 health care workers (HCWs) who have received 2 doses of the BNT162b2 vaccinations. The authentic virus neutralization assay, the surrogate virus neutralizing antibody test (sVNT), and the Anti-N SARS-CoV-2 IgG assay (N-IgG) have been performed along with the S-IgG. The S-IgG correlated well with the neutralizing activity detected by the authentic virus neutralization assay (0.8904. of Spearman's rho value, *p* < 0.0001) and sVNT (0.9206. of Spearman's rho value, *p* < 0.0001). However, 4 samples (2.3%) of S-IgG and 8 samples (4.5%) of sVNT were inconsistent with negative results for neutralizing activity of the authentic virus neutralization assay. The kinetics of the SARS-CoV-2 neutralizing antibodies and anti-S IgG in severe cases were faster than the mild cases. All the HCWs elicited anti-S IgG titer after the second vaccination. However, the HCWs with history of COVID-19 or positive N-IgG elicited higher anti-S IgG titers than those who did not have it previously. Furthermore, it is difficult to predict the risk of breakthrough infection from anti-S IgG or sVNT antibody titers in HCWs after the second vaccination. Our data shows that the use of anti-S IgG titers as direct quantitative markers of neutralizing

**Funding:** This research was partially supported by Japan Agency for Medical Research and Development under Grant Number JP20fk0108472 to TN and by Japan Society for the Promotion of Science Grants-in Aid for Scientific Research under Grant Number 22K15675 to ST. The funders had no role in the study design, data collection and analysis, decision to publish, or preparation of the manuscript.

**Competing interests:** The reagent used in this study were partially provided by abbott, but the study was performed by scientifically proper methods without any bias. This does not alter our adherence to PLOS ONE policies on sharing data and materials.

capacity is limited. Thus, antibody tests should be carefully interpreted when used as serological markers for diagnosis, treatment, and prophylaxis of COVID-19.

## Introduction

Coronavirus disease 2019 (COVID-19) is an infectious disease caused by severe acute respiratory syndrome coronavirus 2 (SARS-CoV-2), which is currently an endemic worldwide [1]. Reverse transcription polymerase chain reaction (RT-PCR) is the gold standard test for the diagnosis of SARS-CoV-2 infection [2, 3]. However, false negative results of RT-PCR can be caused by suboptimal primer design, imperfect RNA extraction techniques, or lower volumes of applied virus [4]. Furthermore, because the RNA concentration of SARS-CoV-2 declines from 1–2 weeks after symptom onset, the detection ratio of RT-PCR is also decreased [5]. On the other hand, because seroconversion of SARS-CoV-2 occurs between one to two weeks after symptom onset, the serological antibody tests' detection rate of specific antibody gradually increases [6].

Many of the commercial antibody tests can specifically detect immunoglobulins, such as IgG and IgM, binding against the nucleocapsid (N) protein and the receptor-binding domain (RBD) in the spike (S) protein of SARS-CoV-2 [7]. They are used as an adjunct to RT-PCR for COVID-19 diagnosis [8]. The RBD directly binds to angiotensin-converting enzyme 2 (ACE2), a host cell receptor that mediates attachment of SARS-CoV-2 [9]. Since 90% of the neutralizing activity against SARS-CoV-2 targets RBD [10, 11], anti-RBD antibodies have the potential to neutralize viral entry into cells and could be a marker of protective immune response against SARS-CoV-2 infection [12, 13]. Assays that detect neutralizing activity are recognized as reliable, but authentic virus neutralization assays are restricted to Biosafety Level 3 (BSL3) facilities, and a few are available. Even the surrogate virus neutralization test (sVNT), based on enzyme-linked immunoassay (ELISA), is time-consuming and has low-throughput. Therefore, high-throughput, widely available quantitative S-IgG has been developed.

In this study, we evaluated whether the antibody titers using the Abbott SARS-CoV-2 IgG II Quant assay, an automated chemiluminescent immunoassay for detecting SARS-CoV-2 S specific antibodies, accurately reflects the antibody dynamics and neutralizing activity following SARS-CoV-2 infection and vaccination using sVNT and full-scale virus neutralization assays.

## Material and methods

### Ethics statement

This study complied with all relevant national regulations and institutional policies. It was conducted in accordance with the tenets of the Declaration of Helsinki and was approved by the Institutional Review Board (IRB) at Juntendo University Hospital (IRB # 20–036). The need for informed consent from individual patients was waived because all samples were de-identified in line with the Declaration of Helsinki. Informed consent was obtained from HCWs (IRB # M20-0089-M01).

### Clinical backgrounds

This study was conducted at Juntendo University Hospital in Japan and included a total of 457 blood serum samples from 253 individuals. Three hundred and twenty-five samples were collected from 128 COVID-19 patients confirmed by RT-PCR between March and September

2020. Of the 128 COVID-19 patients, 100 are inpatient and 28 are outpatient. One hundred and thirty-two samples were collected between June and July 2021 from 132 HCWs (Medical doctors: 90, Nurses: 29, Co-medicals: 6, Clerks: 7) who received a second vaccination between March and April 2021. History of SARS-CoV-2 infection was collected from HCWs through a medical history questionnaire. RT-PCR-based molecular testing/confirmation for SARS-CoV-2 was performed using nasopharyngeal specimens by the 2019 Novel Coronavirus Detection Kit (Shimadzu, Kyoto, Japan) [14]. Specific spike protein mutations were detected using the VirSNiP SARS-CoV-2 mutation assay (Roche diagnostics, Rotkreuz, Switzerland) according to the manufacturer's instructions. Real-time PCR analysis was performed on a light cycler system (Roche, California, United States). Patients using immunosuppressive agents for their underlying diseases before contracting COVID-19 were excluded from this study [15].

We categorized SARS-CoV-2 infected patients into mild, moderate, severe, and critical according to the WHO criteria [16]. Mild COVID-19 was defined as respiratory symptoms without evidence of pneumonia or hypoxia, while moderate or severe infection was defined as presence of clinical and radiological evidence of pneumonia. In moderate cases, $SpO_2 \geq 94\%$ was observed on room air, while one of the following was required to identify the severe and critical cases: respiratory rate $> 30$ breaths/min or $SpO_2 < 94\%$ on room air. Additionally, critical illness was defined as respiratory failure, septic shock, and/or multiple organ dysfunction. We then grouped them into Group M, which included mild and moderate cases, and Group S, which included severe and critical cases. Group M patients with a high-risk background were hospitalized and included in the longitudinal assessment study.

## Serologic testing for SARS-CoV-2

Anti-S SARS-CoV-2 IgG II Quant assay (S-IgG) was performed on the Abbott Alinity i platform (Abbott Laboratories, Chicago, IL, USA) according to the manufacturer's instructions. The assay is based on the chemiluminescent microparticle immunoassay (CMIA) for qualitative detection of anti-S IgG in human serum/plasma against the S glycoprotein on the surface of SARS-CoV-2 [17].

Anti-N SARS-CoV-2 IgG assay (N-IgG) targets N protein and was performed on the Abbott Alinity i platform according to the manufacturer's instructions. The assay is based on the CMIA for semi-quantitative assessment of anti-N IgG. The resulting chemiluminescence in relative light units indicates the strength of the response, which reflects each specific antibody present. Results from the quantitative S-IgG are reported as arbitrary units (AU) per milliliter, and values equal to the cutoff of 50 AU/mL or greater were classified as positive [18]. Results from the semi-quantitative N-IgG are reported as index values, and the manufacturer's suggested positive cutoff point of 1.40 was used [19, 20].

## Virus neutralization assay

The authentic virus neutralization assay has been performed as described previously [21]. The SARS-CoV-2 ancestral strain WK-521 (lineage A, GISAID ID: EPI_ISL_408667) was used for the authentic virus neutralization assay which has been performed at the National Institute of Infectious Diseases (NIID) with ethics approval by the medical research ethics committee of NIID for the use of human subjects (#1178). Briefly, serially diluted serum samples (2-fold serial dilutions starting at 1:5 dilution, diluted with high glucose Dulbecco's Modified Eagle Medium supplemented with 2% Fetal Bovine Serum and 100 U/mL penicillin/streptomycin, from Fujifilm Wako Pure Chemicals, Japan) were mixed with the virus from 100 Median Tissue Culture Infectious Dose (TCID50) and incubated at 37˚C for 1 hour. The mixture was subsequently incubated with VeroE6/TMPRSS2 cells (JCRB1819, JCRB Cell Bank, Japan) and

seeded in 96-well flat-bottom plates for 4–6 days at 37˚C in a chamber supplied with 5% $CO_2$. The cells were then fixed with 20% formalin (Fujifilm Wako Pure Chemicals) and stained with crystal violet solution (Sigma-Aldrich, St Louis, MO). Each sample was assayed in 2–4 wells and the average cut-off dilution index of > 50% cytopathic effect was presented as a neutralizing titer. Neutralizing titer of the sample below the detection limit (1:5 dilution) was set as 2.5. Neutralizing antibody titer of < 5 is considered negative and ≥ 5 was considered positive.

The surrogate virus neutralizing antibody test (sVNT) has been performed using the GenScript cPass® SARS-CoV-2 Antibody Detection Kit, a blocking enzyme-linked immunosorbent assay (GenScript, Piscataway, New Jersey, USA), following the company's instructions. Briefly, the samples and controls were pre-incubated with the horseradish peroxidase (HRP)-labeled recombinant RBD proteins and the mixture was added to a capture plate pre-coated with the hACE2 proteins. After the complex of neutralizing antibody binding RBD-HRP was removed by washing, the wells were read at 450 nm in a microtiter plate reader. The percent signal inhibition for the detection of neutralizing antibodies were calculated as follows:

$$\% \text{ Signal Inhibition} = (1\text{-OD value of Sample/OD value of Negative Control}) \times 100\% \text{ (cutoff value : 30\% signal inhibition)}.$$

## Statistical analysis

Data analysis was carried out using GraphPad Prism software (version 9.0.1; San Diego, CA, USA) and R software (version 4.1.0). Titers of antibodies were log-transformed before statistical analyses. Analysis between antibody titer and neutralization test was performed using the Spearman's rank correlations coefficient.

When analyzing statistical differences between two or more experimental groups, one-way analysis of variance (ANOVA) and Tukey multiple comparison post hoc analysis were used. The Wilcoxon signed rank test was used in the test analysis for the two experimental groups. The following notation was used to show statistical significance: * $p$ value < 0.01, and ** $p$ value < 0.001.

For longitudinal assessment, the kinetics in the emergence of anti-S IgG and sVNT titers were determined for the S (severe and critical) and M (mild and moderate) groups using a nonlinear mixed effects model. The models were fitted to a four-parameter logistic function, with a constrained lower asymptote set to the limit of detection, the inflection point, a scale parameter, and the upper asymptote for Group S and Group M. A comparison between Group S and Group M was conducted in a Z test using the estimations.

## Results

### Correlations between anti-S IgG titer and neutralizing activities

We first compared the results of S-IgG to sVNT and then S-IgG to the neutralizing activity. As shown in Fig 1, the results of 176 samples demonstrated a strong linear correlation between S-IgG and sVNT. The relationship between S-IgG and the authentic virus neutralizing assay also showed a liner correlation. We further confirmed a linear correlation between sVNT and the authentic virus neutralizing assay. Comparison of S-IgG and the authentic virus neutralizing assay revealed a positive percent agreement (PPA) of 96.8% and a negative percent agreement (NPA) of 92.3%. Similarly, a comparison of sVNT and the authentic virus neutralizing assay revealed a PPA of 97.6% and a NPA of 84.6%. However, when compared to the authentic virus neutralization assay, 4 samples (2.3%) tested for S-IgG and 8 samples (4.5%) tested for sVNT were inconsistent with the negative results for the neutralizing activity of the

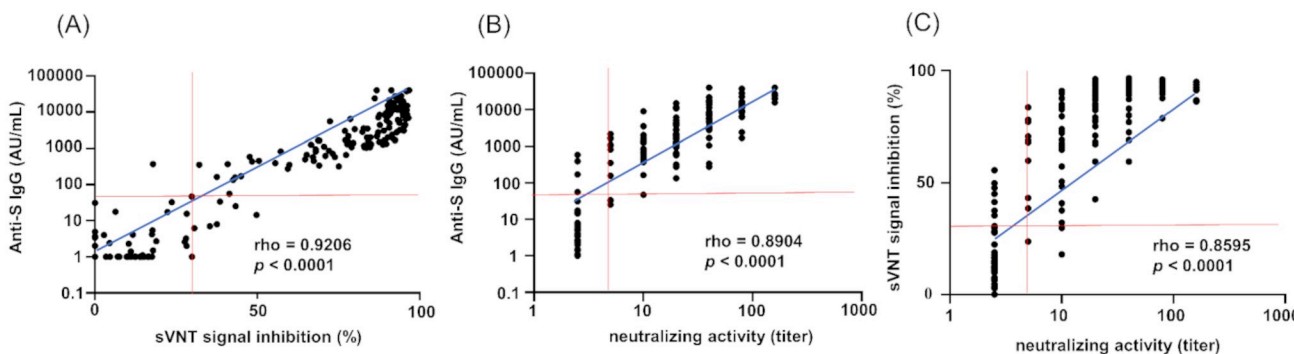

**Fig 1. Comparison of S-IgG with sVNT and authentic virus neutralization assay.** (A) Correlation between anti-S IgG and sVNT titers. Spearman's rank correlations coefficient (rho) value 0.9206, p < 0.0001, 95% CI (0.8937 ~ 0.9410). (B) Correlation between anti-S IgG and neutralizing activity titers. Spearman's rank correlations coefficient (rho) value 0.8904, p < 0.0001, 95% CI (0.8540 ~ 0.9182). (C) Correlation between sVNT and neutralizing activity titers. Spearman's rank correlations coefficient (rho) value 0.8595, p < 0.0001, 95% CI (0.8137 ~ 0.8946). The horizontal axis and the vertical axis are in logarithmic notations.

comparative method. On the other hand, 4 samples (2.3%) tested for S-IgG and 3 samples (1.7%) tested for sVNT were inconsistent with the positive results for the neutralizing activity of the comparative method. Samples with a titer $\geq 20$ in the authentic virus neutralizing assay were all positive in S-IgG and sVNT.

## Longitudinal assessment of antibody titers in COVID-19 patients

To examine chronological changes in anti-S IgG and sVNT titers, we plotted these titers of inpatients whose antibodies were measured three or more times. Samples were collected up to 60 days after symptom onset to determine the antibodies' rate of change. The age distribution of each group was 50–90 years for Group S and 20–80 years for the Group M. Anti-S IgG and sVNT titers from patients of Group S and Group M were plotted against time from symptom onset and fitted (Fig 2).

Group S showed earlier increases in both anti-S IgG and sVNT titers than Group M. The 80% maximal response of anti-S IgG was achieved on day 17 for Group S and on day 23 for Group M. Similar kinetics were observed with sVNT, which achieved the 80% maximal response in Group S on day 15 and in Group M on day 23. No significant difference of maximal plateau value between Group S and Group M was observed for both anti-S IgG and sVNT (S-IgG: Group S, 7617.0, Group M, 4131.9; sVNT: Group S, 91.9, Group M, 91.0).

## Distribution of anti-S IgG and sVNT titers after second vaccination

We then investigated S-IgG, sVNT, and N-IgG in 132 HCWs who received two doses of the vaccine. Because seropositive individuals with N-IgG are considered as previously infected with SARS-CoV-2 regardless of symptoms, the tested individuals were divided into four groups based on N-IgG results (positive/negative) and COVID-19 medical history. We observed that all tested individuals were seropositive with both S-IgG and sVNT. As shown in Fig 3, anti-S IgG and sVNT titers in N-IgG negative individuals with no medical history of COVID-19 were significantly lower compared to those with COVID-19 medical history and/or N-IgG positive. The N-IgG positive individuals showed comparable anti-S IgG and sVNT titers to those with COVID-19 medical history.

Among these HCWs, we assessed the anti-S IgG and sVNT titers in breakthrough infection cases who were diagnosed with SARS-CoV-2 infection by RT-PCR after the second

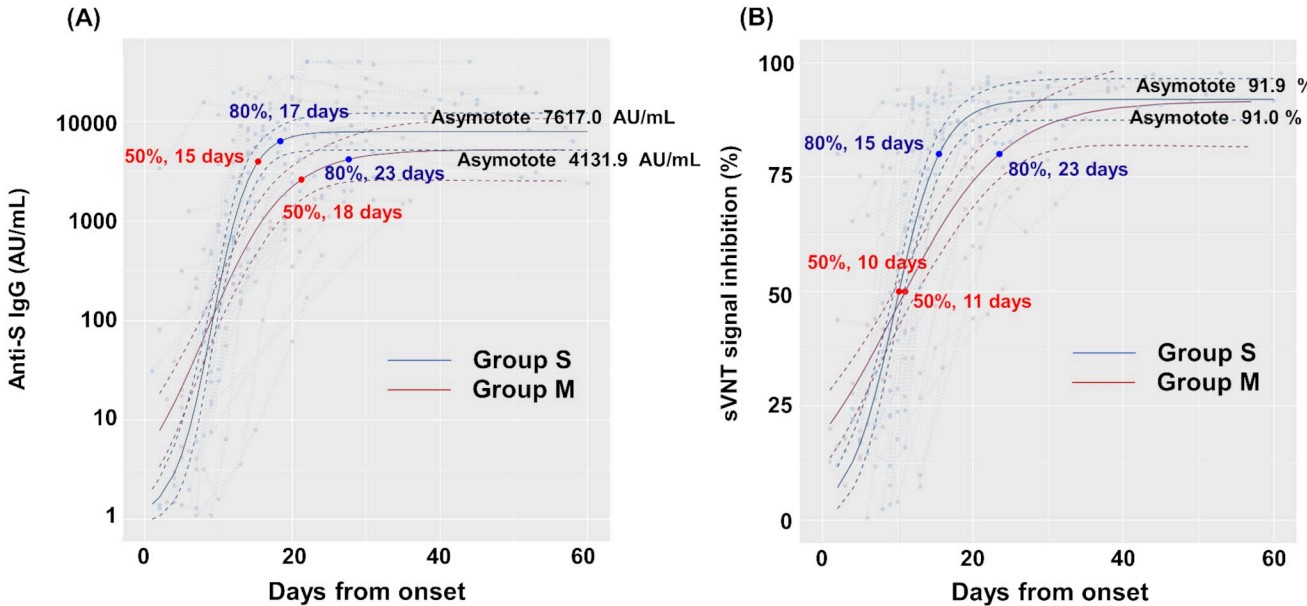

**Fig 2. Longitudinal change of anti-S IgG and sVNT titers.** Anti-S IgG titers (A) and sVNT titers (B) of 253 samples from 63 hospitalized patients (Group S, 134 samples from 27 cases; Group M, 119 samples from 36 cases) were plotted against time from symptom onset and fitted (solid line). Red and blue dots indicate the calculated time required to achieve the 50% and the 80% maximal neutralization titer, respectively. The plateau values for anti-S IgG and sVNT titers of Group S and M individuals were shown (Z test). The vertical axes are in logarithmic notation.

vaccination (n = 19). Of the 19 breakthrough infection cases, 13 were infected with the B.1.617.2 (delta) variant and 6 were unknown during this study period. As shown in Fig 4, no difference of anti-S IgG nor sVNT titers were observed between the breakthrough infection cases and uninfected controls.

## Discussion

In this study, we investigated the serological of antibody dynamics and neutralizing potency following SARS-CoV-2 infection and post vaccination by comparing three quantitative assays with different principles for detection of antibodies to SARS-CoV-2. S-IgG is correlated well with both sVNT and the authentic virus neutralization assay with high PPA and NPA. However, when the authentic virus neutralization assay was used as the comparative method, the results of 2.3% for S-IgG and 4.5% for sVNT were inconsistent with the negative results for neutralizing activity of the comparative method. Similar results have been reported in other studies [22].

These findings demonstrate a discrepancy between serological antibody levels and neutralizing activity detected by authentic virus neutralization assay. The binding capacity of virus-specific IgG antibodies is known to increase over time, termed as affinity maturation [23, 24]. The binding affinity and neutralizing potency of anti-RBD antibodies in SARS-CoV-2 infection have also been reported to increase over time [10, 25]. This process may be associated with a slower increase in the authentic virus neutralization activity compared to the anti-S IgG and sVNT titers [26]. In this study, severe/critical cases of SARS-CoV-2 infection (Group S) showed more rapid evolution of anti-S IgG and sVNT titers than mild/moderate cases (Group M), which was consistent with previous reports [27–29]. Neutralizing antibodies directly block infection, whereas innate and T-cell responses that mediate neutralizing capacity induce

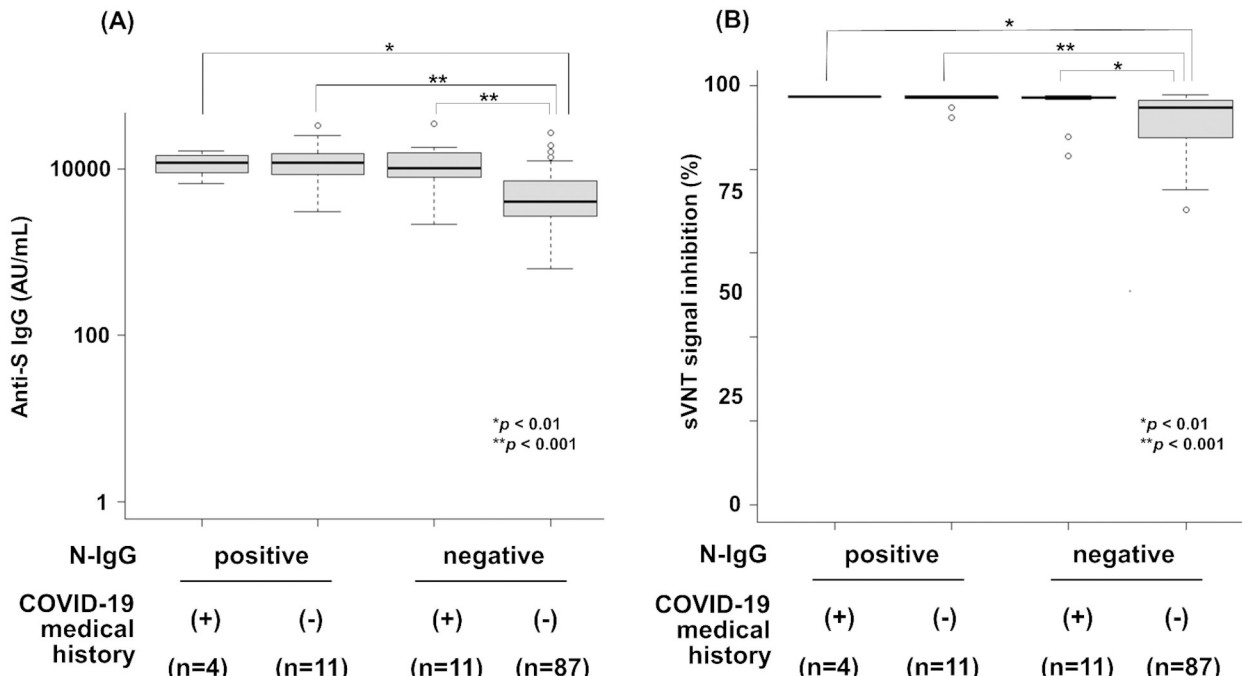

**Fig 3. Distribution of anti-S IgG and sVNT titers in HCWs after second vaccination.** Anti-S IgG titers (A) and sVNT titers (B) were quantified in post-vaccination HCWs (n = 113) including N-IgG positive with COVID-19 medical history (n = 4) or without COVID-19 medical history (n = 11), and N-IgG negative with COVID-19 medical history (n = 11) or without COVID-19 medical history (n = 87). Statistical analysis was performed using one-way ANOVA, and statistical significance is indicated as follows: $^*p < 0.01$, $^{**}p < 0.001$. The median antibody titer and interquartile range (IQR) of the anti-S IgG titer and sVNT titer in each group: N-IgG positive with COVID-19 medical history, 11705 AU/mL (IQR 7705–15329), 97.3% (IQR 97.3–97.4); N-IgG positive without COVID-19 medical history, 11779 AU/mL (IQR 5973–16610), 97.2% (IQR 96.8–97.4); N-IgG negative with COVID-19 medical history, 10220 AU/mL (IQR 7583–16548), 97.1% (IQR 96.6–97.3); and N-IgG negative without COVID-19 medical history, 3961 AU/mL (IQR 2622–7175), 94.6% (IQR 87.5–96.5).

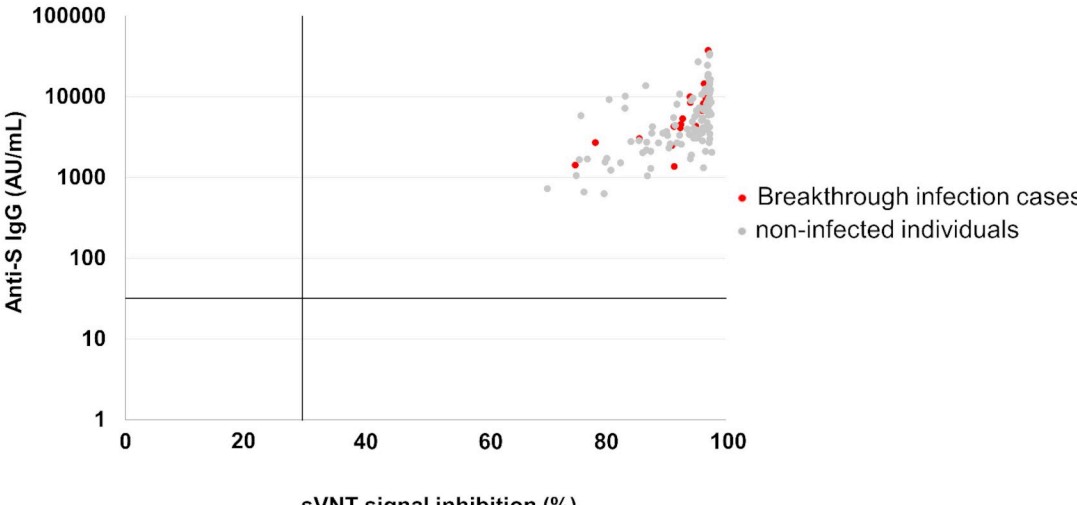

**Fig 4. Anti-S IgG and sVNT titers of the cases with breakthrough infection and non-infected individuals after the second vaccination.** Anti-S IgG and sVNT titers in post-second vaccination HCWs (n = 132) were plotted; 19 samples were from individuals with breakthrough infection at a later date (red dots) and 113 samples were from non-infected individuals (black dots).

hyperactivated inflammation and promote severity [30, 31]. Further studies are required to elucidate whether the rapid induction of anti-S IgG and the persistence of the antibody response contributes to the protection against infection and prevention of severe diseases.

Regarding the role of vaccination and induced humoral immune response after vaccination, we investigated anti-S IgG and sVNT titers in HCWs who received two doses of the BNT162b2 vaccination. A strong correlation between anti-S IgG and sVNT titers was observed in sera approximately 1–2 months after the second vaccination. Significantly higher anti-S IgG and sVNT titers were observed in individuals with a COVID-19 medical history as well as N-IgG positive individuals who have not been diagnosed with COVID-19 due to lack of symptoms.

Of note, no significant difference in both anti-S IgG and sVNT titers were observed in breakthrough infected individuals compared to the ones without infection. Previous studies reported that anti-S IgG and neutralizing antibody titers were inversely related to the increased risk of breakthrough infection [32–34]. On the other hand, several studies demonstrated no significant difference in antibody titers with or without breakthrough infection [35, 36]. The discrepancy of the results may have been affected by the measurement time after vaccination, epidemic variants of SARS-CoV-2, immune status of individuals, and sample scale [35].

The production of anti-S IgG or neutralizing antibodies may be related to various factors, such as age, medications, and underlying diseases [37]. In our study, B.1.617.2 (delta) was the predominant variant in breakthrough infected individuals. Of the 19 breakthrough infection cases, 13 were diagnosed with a delta variant infection and 6 were unknown in this study. The efficacy of two doses of BNT162b2 vaccine has been reported to be 88.0% against the delta variant [38].

Even if a high antibody titer is obtained through vaccination, breakthrough infection from variants of concern (VOC) will become more likely to occur because the ability to target and neutralize the receptor binding motif (RBM) on the S protein of VOCs is reduced [39, 40]. Therefore, anti-S IgG and sVNT titers might not be effective indicators of breakthrough infections.

This study has several limitations. Firstly, it was conducted in a single university hospital with a relatively small number of samples. Secondly, there may be an age bias between Group S and Group M of the COVID-19 cases. Finally, post-vaccination antibody measurements were performed only once, and changes in antibody titers over time could not be followed.

In conclusion, using anti-S IgG titers as direct quantitative markers of neutralizing potency is limited. Therefore, serologic tests need to be carefully interpreted in the treatment of COVID-19. On the other hand, anti-S IgG antibody titers can provide information on antibody acquisition both from past infections and vaccination, suggesting that high-throughput, widely available quantitative S-IgG measurements using the Abbott SARS-CoV-2 IgG II Quant assay can be used in epidemiological studies to provide important information for future SARS-CoV-2 infection control.

## Acknowledgments

The authors thank the Department of Research Support Utilizing Bioresource Bank, Juntendo University Graduate School of Medicine, for use of their facilities.

## Author Contributions

**Conceptualization:** Kazuhisa Takahashi, Toshio Naito, Yoko Tabe.

**Data curation:** Shuko Nojiri.

**Funding acquisition:** Satomi Takei.

**Investigation:** Kaori Saito, Maika Takahashi, Gene Igawa, Takamasa Yamamoto, Abdullah Khasawneh, Faith Jessica Paran, Yuki Horiuchi, Mitsuru Wakita.

**Methodology:** Takayuki Kanno, Minoru Tobiume, Tadaki Suzuki.

**Resources:** Makoto Hiki, Atsushi Okuzawa.

**Supervision:** Takashi Miida.

**Writing – original draft:** Toshihiro Takahashi.

**Writing – review & editing:** Tomohiko Ai.

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
