## [Decision Letter · Decision Letter 0]

7 Jun 2023

PONE-D-22-35551Assessment of antibody dynamics and neutralizing activity using serological assay after SARS-CoV-2 infection and vaccinationPLOS ONE

Dear Dr. Tabe,

Thank you for submitting your manuscript to PLOS ONE. After careful consideration, we feel that it has merit but does not fully meet PLOS ONE’s publication criteria as it currently stands. Therefore, we invite you to submit a revised version of the manuscript that addresses the points raised during the review process. The methods are important for this journal.We have to go to the second round of the review process.

We look forward to receiving your revised manuscript.

Kind regards,

Etsuro Ito

Academic Editor

PLOS ONE

https://www.nature.com/articles/s41598-022-19073-z?code=86adafd0-3d6f-4c5a-a73e-8355546cc298&error=cookies_not_supported

In your revision ensure you cite all your sources (including your own works), and quote or rephrase any duplicated text outside the methods section. Further consideration is dependent on these concerns being addressed.

“This research was partially supported by Japan Agency for Medical Research and Development under Grant Number JP20fk0108472 to TN and by Japan Society for the Promotion of Science Grants-in Aid for Scientific Research under Grant Number 22K15675 to ST.”

“The reagent used in this study were partially provided by abbott, but the study was performed by scientifically proper methods without any bias.”

Reviewers' comments:

Reviewer's Responses to Questions

**Comments to the Author**

1. Is the manuscript technically sound, and do the data support the conclusions?

Reviewer #1: Yes

Reviewer #2: Partly

2. Has the statistical analysis been performed appropriately and rigorously? 

Reviewer #1: Yes

Reviewer #2: Yes

3. Have the authors made all data underlying the findings in their manuscript fully available?

Reviewer #1: Yes

Reviewer #2: Yes

4. Is the manuscript presented in an intelligible fashion and written in standard English?

Reviewer #1: Yes

Reviewer #2: Yes

5. Review Comments to the Author

Reviewer #1: [General Comments]

The authors showed that antibody dynamics and neutralizing activity following COVID-19 vaccination/ infection among patients, HCWs. Although the study poluation and back ground seems unique, there're some points to be revised. And the manuscript is redundunt with too many figures. Simplify is also required.

[Specific Comments]

1. Introduction

The PECO of the study is unclear. 'Investigate the serological kinetics' seems to be vague.

2. Method

Line 105; Should clalify the number of community acquired/ nosocomial COVID-19 case.

Line 106; Should describe the breakdown of HCWs (i.e. Nurses N= , doctors N=, Lab technician N= ...)

3. Result

The first half of result is overlapped with table1. Table 1 should be deleted. Or simplify the manuscript including quotation of Table1.

For Figure 2A, Plot of each group (S and M) should combined.

For Figure 2B, Plot of each group (S and M) should combined.

Number of the categolized study population shoud be noted on the figure. (i.e. number of the subject in N-IgG pisitive & COVID-19 Medical History positive in Figure 3A.)

Most of figure legends are overlapped with each figures. Delete overlapped description in the manuscript.

4. Discussion

The authors add the thought, or plan, of potential for clinical use in real world according to the result of the study.

Reviewer #2: PONE-D-22-35551_reviewer

This is an important study to compare the three different antibody analysis methods and their kinetics among COVID-19 cases and HCWs after 2-dose vaccination, using authentic virus neutralization assay as a reference. Please consider to revise the manuscript, based on the comments listed below.

Major comments

Abstract

L 47-49: Does this sentence mean that “low titer of” anti-S IgG or sVNT causes breakthrough infection? The meaning is a little bit difficult to understand.

Overall, please consider to revise the abstract accordant with the revision of the main text, tables and figures.

Introduction

The aim of this study is not matched with the conclusion. Please revise in accordance with the other contents of the manuscript.

Material and Methods

L 109-110: Immunosuppressive agents are commonly used to treat COVID-19 itself. Does this study exclude COVID-19 patients treated with immunosuppressive agents such as corticosteroid or other biologic agents? Or are the patients using immunosuppressive agents for their underlying diseases before contracting COVID-19 excluded ?

It is unclear how the authors collected the history of contracting COVID-19 from the HCWs and how they identified infection with delta variant or other variants.

L120: Please describe the detailed material and methods of the long-term evaluation study.

L170: The meaning of “When experiments involved more than two groups” is difficult to understand.

L175: What is the longitudinal analysis? Is it same as “long-term evaluation study”? Please clarify the meaning.

Results

Overall, description about the methods seems mixed in the Result part. For example, L182-183, L214-221, L240-245.

L269-272: As pointed out in the Material and Methods part, please clarify the methods to identify the past history of COVID-19 of HCWs and type of variants, and revise the description if necessary.

Discussion

L294-296: Relatedness with the former sentences in L292-294 is a little bit confusing. Do the authors intend to say the authentic virus neutralization activity increases more slowly/weakly than anti-S antibody and/or sVNT?

L328: the distribution of age in the Groups should be written in the Result part.

L331-333: The conclusion is not matched with the aim of the study described at the end of the Introduction part (investigation or serologic kinetics).

The authors conclude that usefulness of titers of anti-S IgG and sVNT as surrogate markers of neutralizing capacity is limited, But as an experiment, concordance of titers of anti-S IgG and sVNT with authentic neutralization assay seems acceptable, even though some cases show discrepancy. Do the authors intend to describe the insufficiency of the three markers to predict the ability to prevent infection?

Minor comments

L114: SpO2 should be written with small "2".

L133: AU/ml should be written as AU/mL.

6. PLOS authors have the option to publish the peer review history of their article (what does this mean?). If published, this will include your full peer review and any attached files.

Reviewer #1: No

Reviewer #2: No

---

## [Author Response · Author response to Decision Letter 0]

16 Aug 2023

August 15th, 2023

Dr. Etsuro Ito

Academic Editor

PLOS ONE

RE: Resubmission of the revised manuscript MS# PONE-D-22-35551R1 titled “Assessment of antibody dynamics and neutralizing activity using serological assay after SARS-CoV-2 infection and vaccination” by Takahashi T., et al.

Dear Dr. Etsuro Ito,

We appreciate the reviewers for their highly constructive comments for our manuscript, and we believe that we fully addressed the issues as detailed below.

Reply (Answer; A) to Reviewer’s comments and questions (Comment; C):

Reviewer 1

COMMENTS by Reviewer #1: 

The authors showed that antibody dynamics and neutralizing activity following COVID-19 vaccination/ infection among patients, HCWs. Although the study poluation and background seems unique, there're some points to be revised. And the manuscript is redundant with too many figures. Simplify is also required.

C1: The PECO of the study is unclear. 'Investigate the serological kinetics' seems to be vague.

A1: We appreciate the reviewer’s suggestion. Following the comment, we rephrased the sentence. (L74-77)

C2: In Line 105, should clarify the number of community acquired/ nosocomial COVID-19 case. In Line 106, should describe the breakdown of HCWs (i.e., Nurses N=, doctors N=, Lab technician N= ...)

A2: Following the reviewer’s suggestion, we added the detail number of patients and HCWs. (L92,94)

C3: 

a. The first half of result is overlapped with Table1. Table 1 should be deleted. Or simplify the manuscript including quotation of Table1.

b. For Figure 2A, Plot of each group (S and M) should combined. 

For Figure 2B, Plot of each group (S and M) should combined.

c. Number of the categolized study population shoud be noted on the figure. (i.e., number of the subject in N-IgG pisitive & COVID-19 Medical History positive in Figure 3A.)

d. Most of figure legends are overlapped with each figures. Delete overlapped description in the manuscript.

A3: Thank you for the valuable comments. Following the reviewer’s suggestion, we changed as follows.

a. Table1 has been deleted. 

b. We combined the plot of each group (S and M) and changed the legend accordingly in Figure 2A and B.

c. We added the number of the categorized study population in Figure 3A.

d. We deleted the description in the manuscript and figure which were overlapped with figure legends for Figure 1 and 2. 

C4: The authors add the thought, or plan, of potential for clinical use in real world according to the result of the study.

A4: We appreciate the reviewer's suggestion. A description of the potential for medical epidemiological use in the real world has been added. (L307-313)

Reviewer 2

COMMENTS by Reviewer #2: 

This is an important study to compare the three different antibody analysis methods and their kinetics among COVID-19 cases and HCWs after 2-dose vaccination, using authentic virus neutralization assay as a reference. Please consider to revise the manuscript, based on the comments listed below.

C1: L 47-49: Does this sentence mean that “low titer of” anti-S IgG or sVNT causes breakthrough infection? The meaning is a little bit difficult to understand.

Overall, please consider to revise the abstract accordant with the revision of the main text, tables and figures.

A1: We apologize for the confusion caused by the previous explanation. We described that it is difficult to determine whether an antibody titer of anti-S IgG or sVNT is causing a breakthrough infection (L46-47). The overall correction has been made to reflect other comments. 

C2: The aim of this study is not matched with the conclusion. Please revise in accordance with the other contents of the manuscript.

A2: Following the reviewer’s suggestion, the description of the aim has been changed. (Abstract, L29-38; introduction, L74-77)

C3:

a. L 109-110: Immunosuppressive agents are commonly used to treat COVID-19 itself. Does this study exclude COVID-19 patients treated with immunosuppressive agents such as corticosteroid or other biologic agents? Or are the patients using immunosuppressive agents for their underlying diseases before contracting COVID-19 excluded ?

b. It is unclear how the authors collected the history of contracting COVID-19 from the HCWs and how they identified infection with delta variant or other variants.

c. L120: Please describe the detailed material and methods of the long-term evaluation study.

d. L170: The meaning of “When experiments involved more than two groups” is difficult to understand.

e. L175: What is the longitudinal analysis? Is it same as “long-term evaluation study”? Please clarify the meaning.

A3:

a. We apologize for any confusion caused by the previous description. We clarified that in this study, we excluded patients who had received immunosuppressive drugs for their underlying disease prior to contracting COVID-19 (L101-102).

b. We apologize for the inadequate description. We described how we identified the past history of COVID-19 infection in HCWs and identified the type of variants (L95-96, L98-101).

c. We described the analytical method of the longitudinal assessment study in the Statistical analysis section in “Material and Methods” (L167-169).

d. We apologize for the unclear wording. We corrected the sentences. (L162-164)

e. We reworded “long-term evaluation study” to “longitudinal assessment study” (L112,167).

C4: 

a. Overall, description about the methods seems mixed in the Result part. For example, L182-183, L214-221, L240-245.

b. L269-272: As pointed out in the Material and Methods part, please clarify the methods to identify the past history of COVID-19 of HCWs and type of variants and revise the description if necessary.

A4: 

a. Thank you for bringing this to our attention. We apologize for the mix up. Following the reviewer’s suggestion, we have deleted the duplicate text in the "Results" section.

b. Please see A3b.

C5:

a. L294-296: Relatedness with the former sentences in L292-294 is a little bit confusing. Do the authors intend to say the authentic virus neutralization activity increases more slowly/weakly than anti-S antibody and/or sVNT?

b. L328: the distribution of age in the Groups should be written in the Result part.

c. L331-333: The conclusion is not matched with the aim of the study described at the end of the Introduction part (investigation or serologic kinetics).

d. The authors conclude that usefulness of titers of anti-S IgG and sVNT as surrogate markers of neutralizing capacity is limited, but as an experiment, concordance of titers of anti-S IgG and sVNT with authentic neutralization assay seems acceptable, even though some cases show discrepancy. Do the authors intend to describe the insufficiency of the three markers to predict the ability to prevent infection?

A5: 

a. Thank you for your helpful comment. We rephased the sentence. (L272-274)

b. Following the reviewer’s suggestion, the age distribution was noted in the "Results" section and the statement of the discussion was changed. (L203-204, L304-305)

c. We agree with the reviewer’s comment and changed the statements of the aim and the conclusion. Please see A2.

d. Yes, we conclude that the use of anti-S IgG titers as a direct quantitative marker of neutralizing capacity has limitation and that serological tests should be interpreted with caution in the treatment of COVID-19. On the other hand, we also noted that quantitative S-IgG measurements suggest that they can be used in epidemiological studies to provide important information for future control of SARS-CoV-2 infection. (L307-313)

COMMENTS (continued):

The authors might consider some minor comments:

C1: L114: SpO2 should be written with small "2".

L133: AU/ml should be written as AU/mL.

A1: Corrected.

---

## [Decision Letter · Decision Letter 1]

4 Sep 2023

Assessment of antibody dynamics and neutralizing activity using serological assay after SARS-CoV-2 infection and vaccination

PONE-D-22-35551R1

Dear Dr. Tabe,

We’re pleased to inform you that your manuscript has been judged scientifically suitable for publication and will be formally accepted for publication once it meets all outstanding technical requirements.

Kind regards,

Etsuro Ito, Ph.D.

Academic Editor

PLOS ONE

Reviewers' comments:

Reviewer's Responses to Questions

**Comments to the Author**

1. If the authors have adequately addressed your comments raised in a previous round of review and you feel that this manuscript is now acceptable for publication, you may indicate that here to bypass the “Comments to the Author” section, enter your conflict of interest statement in the “Confidential to Editor” section, and submit your "Accept" recommendation.

Reviewer #1: All comments have been addressed

Reviewer #2: All comments have been addressed

2. Is the manuscript technically sound, and do the data support the conclusions?

Reviewer #1: Yes

Reviewer #2: Yes

3. Has the statistical analysis been performed appropriately and rigorously? 

Reviewer #1: Yes

Reviewer #2: Yes

4. Have the authors made all data underlying the findings in their manuscript fully available?

Reviewer #1: Yes

Reviewer #2: Yes

5. Is the manuscript presented in an intelligible fashion and written in standard English?

Reviewer #1: Yes

Reviewer #2: Yes

6. Review Comments to the Author

Reviewer #1: The authors revised well according to the recommendations. The authors should conduct similar investigations.

Reviewer #2: The authors revised the manuscript following the previous suggestions and comments appropriately, and made the manuscript more clearly understandable.

7. PLOS authors have the option to publish the peer review history of their article (what does this mean?). If published, this will include your full peer review and any attached files.

Reviewer #1: **Yes: **Yuji Hirai

Reviewer #2: No

---

## [Editor Report · Acceptance letter]

11 Sep 2023

PONE-D-22-35551R1 

Assessment of antibody dynamics and neutralizing activity using serological assay after SARS-CoV-2 infection and vaccination 

Dear Dr. Tabe:

I'm pleased to inform you that your manuscript has been deemed suitable for publication in PLOS ONE. Congratulations! Your manuscript is now with our production department. 

Kind regards, 

on behalf of

Prof. Etsuro Ito 

Academic Editor

PLOS ONE